# Reception Operators' Perception of the Labor Market Integration of Refugees in Light of the Social Cognitive Career Theory

**Diego Boerchi**

Faculty of Education, Department of Psychology, Università Cattolica del Sacro Cuore, 20123 Milano, Italy; diego.boerchi@unicatt.it

**Abstract:** The millions of refugees living abroad are supported by reception operators in integrating into the hosting country's labor market. Because the operators are usually not experts in career counseling, it is essential to investigate how they act and consequently interpret their role in supporting the labor integration process, which could, at least in part, explain the weaknesses in the migrants' labor situation. The study is based on fourteen narrations from reception operators on migrants whom they have followed for a career-counseling intervention. The Social Cognitive Career Theory has been the theoretical reference both to define the hypotheses to be tested and the coding of the narratives. The main result is that operators tend to contribute in the final part of the process, i.e., when the migrants have to identify their professional goals and choose which actions to take to achieve them. Furthermore, they view their role as prescriptive and substitutive, not as supporting self-awareness, contextual knowledge, and the choice process. This can produce immediate actions of finding employment, often of low quality, rather than developing interests and projections towards a career that creates satisfaction, and adverse reactions in qualified migrants.

**Keywords:** immigrants; refugees; career development; career adjustment; expectations; career counseling; career education

## 1. Introduction

At the end of 2021, because of persecution, conflict, violence, human rights violations, or events alarming public order, 89.3 million people were forced to be displaced worldwide, 31.7 million abroad as refugees or asylum seekers (UNHCR 2021). The integration process in the host country is consistently connected with their occupation. Securing a qualitative and stable job position is crucial for their integration because it is considered a signal of being accepted and acknowledged by the host society (Sniderman et al. 2004) and can promote cultural adjustment, interethnic relations, and norms sharing (Vollebergh et al. 2003). Unfortunately, many refugees and asylum seekers struggle to integrate successfully into the host country's labor market, incurring poorer outcomes when compared to the native population and other migrants (Aydemir 2011; Connor 2010; Wilkinson 2008).

Some of them must accept jobs for which they are overqualified, which produces work stress and dissatisfaction (Chan et al. 2016). Others focus on low-skilled jobs, pushed by the urgency of earning enough to survive, resulting in labor market segregation (McDowell et al. 2009) and concentrating on low-quality and precarious jobs. It is pretty standard for the existence of ethnic enclaves (Heisler 2000) based on networks which, for the limited quality of connections they provide, are helpful in assisting migrants with finding a job in the short term, but they also enforce further segmentation (Venturini and Villosio 2018) and prevent upward mobility (Fuller and Martin 2012; Seibel and van Tubergen 2013).

Insufficient attention to credentials, skills, and work experiences in the host country (Baranik et al. 2018; Krahn et al. 2000) and the motivations and biases of employers (Coates and Carr 2005), more focused on a generic and stereotyped perception of candidates as

suitable and motivated only for low-skilled jobs, reduce over time migrants' self-esteem and self-efficacy perception (Willott and Stevenson 2013), and loss of motivation in education (Dancygier and Laitin 2014) and career expectations (Akkaymak 2017).

According to a systematic review of the literature (Morici et al. 2022), most studies concerning migrants' labor market integration center around the Human Capital approach. It is a multi-componential construct focused on individual components, such as knowledge, skills, and personal features, and social elements, such as formal and informal support in finding a job (Becker 1975). The review has defined three main layers of factors that affect the migrants' integration into the labor market. These can be clustered around three main layers: individual (personal and social resources); organizational (employers, employers' organizations, and inclusion services); structural (broader national or international context, including the laws and regulations on credential accreditation and permission to work in the host country). Studies based on the most typical psychological career models, on the opposite, are under-represented. The Career Construction Theory (CCT) (Savickas 2013; Savickas and Porfeli 2012) could be very useful in explaining how personal features and social context should contribute to shaping more effective and satisfactory career projects in refugees. More unexpected is the lack of studies based on the Social Cognitive Career Theory (SCCT) (Lent et al. 2003), a model widely considered in the scientific research on career development and job satisfaction: "SCCT may be especially valuable in understanding the career development processes of the recent immigrants and refugees. Personal and environmental factors significantly influence their confidence in obtaining employment and growing in a career path of their choice" (Yakushko et al. 2008). The model is an extension of Bandura's (Bandura 1986) Socio-Cognitive Theory. It is aimed at explaining how people: (a) develop academic and career interests, (b) make and modify their training and career choice plans, and (c) achieve the performance of various qualities in their chosen academic and career paths (Lent et al. 2003). The interrogation of the SCOPUS database returns a list of 920 scientific articles mentioning SCCT in the title or abstract. Only three are related to migrants or refugees, and all of them focus on the United States. Moreover, the SCCT has been considered a framework for the final discussion, not a framework for the data analysis: Baran et al. (2018) analyzed interviews employing grounded theory; Codell et al. (2011) focused on the role of sex, age, education level, English proficiency, and the number of years spent as a refugee on the decreasing in the ability to secure meaningful employment; Yakushko et al. (2008) performed a review, reading part of the results in light of the SCCT. This study aims to explain the dynamics and the limits of the labor market integration of refugees and asylum seekers through the analysis of the narrations produced by the reception operators who supported them through an intervention of career counseling. It is based, on one side, on the Human Capital approach, focusing on individual and organizational layers, and on the other side, on the SCCT, which seems to be the best model to integrate the individual layer, in terms of cognitive perceptions, and the organizational layer, in terms of social support or barrier. More specifically, the study intends to focus on environmental, not personal, factors and, more specifically, on the role of reception operators. Suppose it is true that integration into the labor market does not depend exclusively on reception operators' interventions. In that case, it is also true that their role is primarily in promoting effective career choice processes and overcoming some barriers posed by the context.

As far as is known, it is the first time that the SCCT is used not as a general reference but as an interpretative model, both in the definition of specific hypotheses to be tested and in the encoding of the research data available. The hypotheses concern the steps in identifying professional objectives, which, according to the SCCT, can produce positive results in the process of integration into the labor market only if managed in a specific way. Added to their verification, referring to the methodology of the grounded theory, is the attempt to identify an operational model shared by most reception operators, which could explain the critical issues that may emerge from the possible confirmation of the hypotheses.

*Hypotheses Suggested by the Social Cognitive Career Theory*

The SCCT is explained and graphically represented (Figure 1) starting from the original manuscript by Lent et al. (1994), which continues to be the primary reference to describe the model. To help us understand it, we can refer to the choice that students with a migratory background may make regarding their next course of study. Going backward, we can highlight several elements that help define the choice until we find the role played by ethnicity.

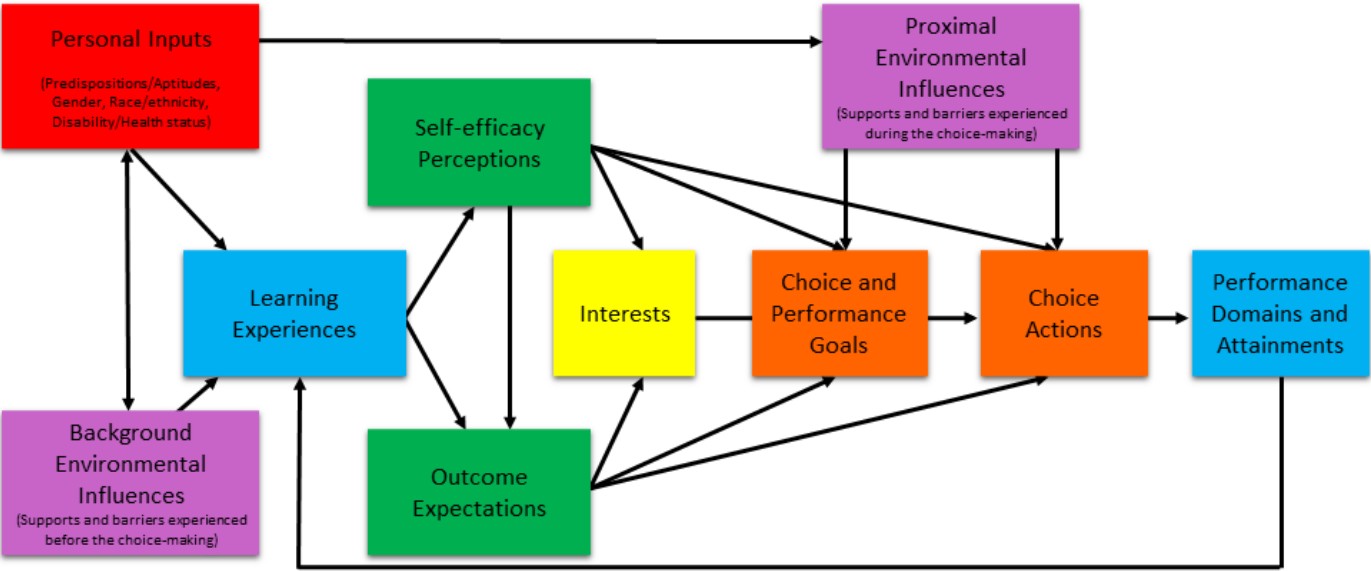

**Figure 1.** Social Career Cognitive Theory (Lent et al. 1994).

Considering that, according to the model, the "attainments" are reached only if the entire process is managed appropriately, each element of the SCCT, which will be described here, will suggest the hypotheses to be tested with regard to the wrong way to behave in order to be effective in defining and reaching professional goals.

<u>Goals and Actions</u>. Goals are the intentions to perform a specific or series of actions and are expected to be the closest antecedents to performing the behavior. About the SCCT model, professional or career interests are considered the most direct antecedents of career goals. In other words, actions depend directly on the goals a person gives himself, and regarding one's career, goals rely directly on the interests a person has developed. With migrant students, it is essential to identify whether they already exist and what career and educational goals they may have. Vocational guidance interventions should support the students in developing goals that are challenging and, at the same time, consistent with the real possibilities, internal and contextual, on which they can rely.

**Hypothesis 1.** *Goals are not functional to career development in the hosting country.*

**Hypothesis 2.** *Goals are not based on migrants' interests.*

**Hypothesis 3.** *The actions are not functional for achieving work objectives.*

<u>Professional interests</u>. They refer to a person's concern in pursuing certain professions or furthering specific subjects through study. In the SCCT model, the interests depend on self-efficacy beliefs and outcome expectations. Suppose people believe they have the skills to perform tasks related to a specific job and perceive positive outcomes associated with getting that job. In that case, they are more likely to develop an interest in a particular topic of study or occupational field. Beyond the influence of specific environmental factors, which

will be discussed below, it is easy to understand why there is a tendency to replicate work patterns from one's family or ethnic background. It increases the likelihood that students will experience skills in specific tasks and makes visible the results they might achieve, directing them to develop an interest in that field. Vocational guidance interventions should bring to consciousness other skills possessed by the students and reason with them as to what achievements they might make in areas other than the one usually most related to their ethnicity of origin to open spaces for alternative choices.

**Hypothesis 4.** *Professional interests are not valid because they are based on unreliable self-efficacy expectations and/or outcome expectations.*

**Hypothesis 5.** *No professional interests have developed.*

Self-efficacy perceptions. Self-efficacy refers to people's judgments about their abilities to organize and execute courses of action required to achieve defined types of performance. Research has shown that high self-efficacy beliefs relate to several outcomes, including ambition, choice behaviors, and job change intentions. According to the SCCT, predictions relate to self-efficacy in specific tasks. Thus, rather than a general effect of efficacy expectations in predicting future attitudes and behaviors, efficacy expectations related to specific tasks affect specific outcomes, such as career interests, goals chosen, and behaviors. With migrant students, it is essential to identify their general expectations about their educational and career skills but, more importantly, how they evaluate their ability to perform more specific tasks (e.g., the ability to study, to present themselves adequately in the labor market, to overcome barriers they might encounter, and so on). Self-efficacy beliefs, because they are not factual data but "beliefs", develop from the person's substantial experience and from the beliefs' influence on the people with whom they interface. Because of this, they can be modified by a more realistic view of opportunities and can be more functional to a more challenging career path.

**Hypothesis 6.** *Self-efficacy expectations are incomplete or overestimate certain aptitudes and skills.*

Outcome expectations. Outcome expectations concern perceptions of the consequences of acting out specific behaviors. In the SCCT model, behavior is seen as largely dependent on perceptions of the particular outcomes that follow from the behavior and the value placed on those outcomes. Thus, people are more likely to choose a specific academic or vocational path if they see the possibility that they can obtain favorable results from their behavior. With migrant students, it is crucial to investigate their outcome expectations regarding certain behaviors in their educational and career future. Suppose they are convinced, for example, that despite their efforts in studying, they will never be able to achieve significant results in education and work. In that case, this will significantly influence their interests and choices. The real possibility of being able to achieve remarkable results and obtaining significant achievements in the study they are doing may not be enough to justify specific choices if the students think that, in any case, they will not be able to aspire to career paths different from those, for example, that their family circle or society ascribe to them.

**Hypothesis 7.** *Outcome expectations are incomplete or far from what can be achieved by choosing a specific career development path.*

Limits and Opportunities. They are the absolute limits and opportunities provided by the specific situation lived by the person, not the personal perception. It is important to name them also because people tend to confuse them with barriers and resources, which in this model have a specific and different meaning.
Learning experiences and Performances. Some experiences and performances are the ones that most affect expectations of self-efficacy and, consequently, the rest of the

dimensions. Students who feel capable because they obtain good results and recognition from adults and peers will be more likely to develop interests in specific fields, make more defined choices, and invest more of their energy in them, and, as a result, be more likely to achieve satisfactory results. Some other experiences are fundamental to learning about the context and presuming what will happen from taking a specific path; bad experiences, like receiving unreliable and prejudiced information, can push the students to develop interests and make choices that will result from their expectations. It is essential to investigate and promote the migrant students' experiences, not making the mistake of considering only the school but also all the informal contexts in which they frequent and helping them to learn about themselves and the context.

**Hypothesis 8.** *Learning experiences are insufficient or focused only on certain situations and activities.*

Barriers and Resources. In the SCCT, these must not be confused with the context's absolute limits and opportunities. Some barriers and resources affect choice through the intermediate step of promoting or impeding learning experiences, affecting interests, self-efficacy perceptions, and outcomes expectations; others are more proximal to the choice, directly affecting goals and actions. We are more used to thinking about the proximal environmental influences, but the background environmental influences are usually more effective and difficult to notice. These drive the students' learning experiences, giving them the perception of having developed their interests independently, while parents and teachers have influenced them. Usually, they intervene in the student's choice at a proximal level if they have failed at the distance level.

**Hypothesis 9.** *In the distal stage, refugees are not helped in capitalizing on what they can learn about themselves and their context.*

**Hypothesis 10.** *In the proximal stage, operators tend to replace migrants in choosing objectives and action strategies, and propose training and work opportunities in a prescriptive way.*

Personal inputs. They are personal characteristics that can affect the sense of self-efficacy and outcome expectations through the mediation of learning experiences. Among them, we find ethnicity, but also gender, disabilities or health status, predispositions, or aptitudes. What characterizes these items is that they are not modifiable, whereas how they are considered and affect learning experiences is adjustable. Their influence is linked to real barriers (e.g., language ability) or prejudicial expectations that distort the perception of the student's abilities and potential and contextual real opportunities for students with migrant backgrounds.

## 2. Materials and Methods

The qualitative study was based on a series of narrations from reception operators on migrants they have followed for a career-counseling intervention. The narrations were coded according to the SCCT model.

### 2.1. Participants

Data were composed of 14 cases of refugees or asylum seekers, illustrated by the same number of operators collaborating with different reception centers in Italy. The sample represented the Italian adult non-economic migrants (see Table 1). At the time of the interviews, migrants were generally young, with ages ranging from 19 to 40 (M = 27.64, SD = 6.04), and mainly males (13 over 14). Twelve came from 8 different African countries, one from Syria and one from El Salvador. At the entrance into Italy, their qualification was mainly low: 7 with elementary school, 4 with middle school, 3 with high school; none graduated. They have been in Italy for 3–6 years (M = 3.93, SD = 0.88). The operators were all graduated, usually with a few years of expertise in roles related to migrants in reception

centers (M = 5.28, SD = 4.92), and illustrated cases they directly managed for support in finding a job (see Table 2).

**Table 1.** Cases.

| Case Number | Status | Gender | Age | Nationality | Years in Italy | Qualification at the Entrance into Italy |
|---|---|---|---|---|---|---|
| 1 | Asylum seeker | M | 40 | Guinea | 3 | Elementary school |
| 2 | Asylum seeker | M | 30 | El Salvador | 6 | Elementary school |
| 3 | Refugee | M | 36 | Sudan | 3 | Elementary school |
| 4 | Asylum seeker | M | 19 | Morocco | 3 | Middle school |
| 5 | Asylum seeker | F | 19 | Nigeria | 4 | Middle school |
| 6 | Refugee | M | 30 | Ivory Coast | 4 | High school |
| 7 | Asylum seeker | M | 24 | Senegal | 4 | Elementary school |
| 8 | Asylum seeker | M | 23 | The Gambia | 3 | Elementary school |
| 9 | Refugee | M | 26 | Syria | 3 | High school |
| 10 | Refugee | M | 34 | Senegal | 4 | Elementary school |
| 11 | Asylum seeker | M | 25 | Nigeria | 4 | High school |
| 12 | Asylum seeker | M | 22 | The Gambia | 4 | Elementary school |
| 13 | Asylum seeker | M | 28 | Nigeria | 5 | Middle school |
| 14 | Refugee | M | 31 | Ghana | 5 | Middle school |

**Table 2.** Operators.

| Case Number | Role in the Reception Center | Years of Expertise | Qualification |
|---|---|---|---|
| 1 | Reception center coordinator | 3 | Degree in political science |
| 2 | Employment advisor | 2 | Degree in law |
| 3 | Trainer | 2 | Degree in literature |
| 4 | Trainer | 2 | Degree in psychology |
| 5 | Employment advisor | 9 | Degree in pedagogy |
| 6 | Reception center coordinator | 9 | Degree in psychology |
| 7 | Reception center operator | 3 | Degree in psychology |
| 8 | Reception center operator | 3 | Degree in political science |
| 9 | Social worker | 5 | Degree in political science |
| 10 | Employment advisor | 6 | Degree in pedagogy |
| 11 | Employment advisor | 21 | Degree in sociology |
| 12 | Reception center coordinator | 3 | Degree in pedagogy |
| 13 | Reception center coordinator | 3 | Degree in pedagogy |
| 14 | Employment advisor | 3 | Degree in communication |

*2.2. Procedure*

The study was developed inside the project FAMI ESPoR, which aimed to design and test an intervention of Career Counseling for Refugees (CCfR). Morici et al. (2022) describe the intervention and its efficacy well. On this occasion, we concentrate on the phase of training the reception operators on the career psychological models previously cited and the features of the CCfR intervention. The participants were also trained on the SCCT: it aimed to help them investigate the situation and career-counseling needs of the migrants, and understand the objectives of the intervention as a whole and concerning individual meetings. In the first step, the model was described theoretically and applied to the high school choice, a choice every one of the participants had made in the past. In the second step, the participants discussed a case proposed by the trainer. In the end, the participants were divided into small groups and asked to select a case among those managed by their members and share it with the classroom. This study is based on the analyses of the cases on which the participant had trained and tested their perception of the migrants' situation before and after knowing the SCCT. This is essential, considering that the reception operators shared the cases, not the migrants directly. Hence, they consist of the operators' interpretation of migrant behaviors and motivations and their behaviors in

the job-search process. In addition, the operators were already trained on the SCCT when they were asked to select the cases, but, as they were not yet experts, their choice has been little impacted by this new interpretative tool. Their oral narrations were transcripted by the classroom tutor.

The narrations have been coded and analyzed with the software NVivo. In the first phase, four narratives were coded according to the elements making up the SCCT. Subsequently, the need to use a more precise coding method emerged, and the entries were re-coded by referring to two subcodes, using only the main code for those rare situations in which it was not possible to identify with certainty the most appropriate subcode. Differently from the Constant Comparative Method, which starts from raw data composed of increasingly large codings and which contain the underlying ones, in this case, we started from a model, and the comparison between the cases suggested integrating the primary coding with a more defined sublevel.

In the first part of the Results section, each code and subcode will be described, and their frequencies will be commented on. Following this, we will propose some relations among different dimensions which appear particularly meaningful. The percentages reported refer to rapport with the total of the references coded.

## 3. Results

*Hp. 1, goals are not functional to career development in the hosting country.* Goals have been coded as Goals Development (3,6%, goals the person has reached or is still investing in), Goals Loss (5.6%, goals that have been abandoned), or Goals (1.7%, where it was not possible to deduct whether they were developing or were abandoned). The number of references related to goal loss is impressive, especially if you consider that it covers 10 cases. In some of them, this choice seems to be congruent with concrete limits set by the context, such as "(4) . . . leads him to think that one day he would like to work as a cook, but at the same time he is aware that it is better to continue the bodywork school to succeed to get a job that will allow him to live independently". In other cases, it appears as resistance to accepting the need to review one's projects in light of a changed context, such as "(1) Despite this experience, Aliu continues to insist even with the operator, telling him that he must find him a job as a metal carpenter".

The number of goal loss confirms hypothesis 1 except for the few cases in which the refugees do not accept the advice to adapt their interests to the new training and work opportunities offered by the new context. This phenomenon suggested looking for a psychological dynamic not considered by the SCCT, which is the elaboration of grieving for the loss of one's professional identity. This indicated the opportunity to add a new code to identify those behaviors typical of those who insist on wanting to carry out the profession previously carried out in the country of origin, even in a context where this is much more difficult, if not impossible. It was named "Grieving" according to Fernández-Valera et al. (2019), who, starting from Kübler-Ross "Five Stages of Grief" (denial, anger, bargaining, depression, and acceptance, 1969), affirm that those who lose their jobs go through a grieving process before they are ready to seek new employment. In this case, the migrants did not lose their jobs but their professional identities. This new code was assigned to the 5.0% of the references and covered six cases, confirming the importance of considering these emotional reactions when dealing with migrants: "(9). He argues that due to the course of a study done in his country, it is not right that he should carry out the same jobs as those who have never studied, and he considers himself superior to the other people welcomed at the center."; "(13) He does not accept living in a small town where there are no artistic possibilities".

*Hp. 2, goals are not based on migrants' interests.* None of the references to objectives contain a reference to professional interests. Therefore, we cannot verify the validity or otherwise of this hypothesis, even if it will be underlined later that the lack of references to professional interests, in general, is typical in all narratives, not only regarding their congruence with the objectives.

*Hp. 3, the actions are not functional for achieving work objectives.* Actions have been coded as Actions Development (7.9%, actions the person is still conducting to reach a goal), Actions Loss (8.6%, actions the person has abandoned or not even started), or Actions (3.0%, where it was not possible to deduct whether they were developing or were abandoned). The number of actions coded is high, and the percentage of active and loss is similar. Active actions were mainly related to training, such as "(5) She enrolls in a 40-h professional Cutting/Sewing course". From the loss of the actions, it is clear that the migrants' outcome expectations have changed because the situation has changed or they have changed their perception of the situation, such as "(12) Pending the outcome of the commission, Simon, unfortunately, loses the motivation to participate in both school and volunteer activities".

It is difficult to say that hypothesis 3 is confirmed because the actions are not functional for achieving work objectives. Even when given up, they are almost always abandoned together with the objectives. Once again, the problem seems to lie in the difficulty in defining realistic goals, as well as respect for the characteristics of the person.

*Hp. 4, professional interests are not valid because they are based on unreliable self-efficacy expectations and/or outcome expectations, and Hp. 5, no professional interests have developed.*

Professional interests have been coded as Interest Development (1.0%, interests in which the person is investing with goals or actions), Interests Loss (2.0%, interests abandoned or lack of interests), or Interests (1.0%, where it was not possible to deduct if they were still active). It is impressive to note that operators very rarely focus on the interests of the migrants (they were cited only in two cases). Interests tended to emerge more as a consequence of a recent experience promoted by the operators, such as "(6) In the account of his experiences, emerges a strong interest in helping others and that he enjoys working in the community of adolescents, especially when he has to help them with their homework and assistance". Interests are lost because of limits of the hosting countries, such as "(13) . . . he loved to take pictures, he made paintings . . . ", while their lack is cited only if directly investigated from the operators: "(11) From the guidance intervention he did not express particular interests".

Also, in this case, it is difficult to say if hypothesis 4 is confirmed and not hypothesis 5, because the lack of expression of professional interests does not mean that they do not exist: it may be that they are only hidden in the belief that manifesting them could be counterproductive for obtaining a job. Conversely, it is clear that refugees are not at the center of interest of both operators and migrants themselves. Considering professional interests are at the center of the SCCT as an element of passage between the current situation (personal and contextual) and the future (objectives, strategies of action, and attainments), it is as if there is an obstacle that separates the two parts, weakening both professional planning and the probability of achieving satisfactory results.

*Hp. 6, self-efficacy expectations are incomplete or overestimate certain aptitudes and skills.* Self-efficacy perceptions have been coded as Congruent self-efficacy perceptions (1.3%, perceptions confirmed by reaching or following investing in a goal or specific performances) or Inconsistent self-efficacy perceptions (1.7%, which has led to abandoning interests and/or goals). Congruent perceptions emerge mainly as a consequence of a specific experience, such as "(5) . . . her skills in the field of tailoring emerge". The situation is similar for inconsistent perceptions, such as "(3) In this meeting, the first difficulties emerge since in Sudan he used a PC in Arabic, the knowledge of the Italian language is basic but not sufficient, and the knowledge of a single graphics program is not adequate to carry out this work".

We can affirm that hypothesis 6 is confirmed but only weakly because, again, there seems to be a lack of attention toward complete and reliable knowledge of the skills and potential of refugees. This is very significant because it underlines that the starting point is not the person, in his uniqueness, but more likely the "user" as belonging to a target.

*Hp. 7, outcome expectations are incomplete or far from what can be achieved by choosing a specific career development path.* Outcome expectations have been coded as Congruent outcome expectations (4.0%, expectations confirmed by reaching or following the investing

on a goal or specific performances) or Inconsistent outcome expectations (9.2%, which has led to abandonment of interests and/or goals). Contrary to self-efficacy perceptions, the outcome expectations are more frequent. An example of outcome expectations is "(4) is a mature boy, compared to his peers, and this allows him to be aware of the new context in which he lives and to make adequate choices for his future". An example of inconsistent expectations is "(1) none of these [experiences] helped him understand that he could not be a metal carpenter".

Hypothesis 7 is one of the most confirmed, demonstrating, as it is easy to assume, the difficulty for people from a different country to understand the possibilities and limits posed by the new context. At the same time, it confirms what has just been said, namely that operators are less interested in the characteristics of refugees and more in the external context.

*Hp. 8, learning experiences are insufficient or focused only on certain situations and activities.* Learning experiences have been coded as Outcomes expectations' learning experiences (5.6%, experiences potentially helpful in improving knowledge of the context and training and work opportunities) or Self-efficacy perceptions' learning experiences (13.2%, experiences potentially helpful in developing realistic and complete self-efficacy perceptions). The latter is the most cited, but the impression is that they are unlikely to translate into a real improvement in the migrants' perception of self-efficacy. The operators seem to report them more to make the readers, not the migrants, understand the skills possessed, such as "(1) many experiences such as volunteering, theatre, the training course, Italian courses." or "(03) The operator, to evaluate Mohamed's preparation in this field and understand if his experience in Sudan can be spent in Italy, decides to contact a graphic designer to ask him if he can give Mohamed a try". The experiences that likely have improved the migrant's outcome perceptions are mainly occasional and not aimed to this scope, such "(5) . . . this desire arises from the fact that in recent years he has interacted with numerous cultural mediators of her territory".

Also, performances can be related to this hypothesis, and they have been coded as Positive performances (6.9%, where the migrants have obtained good performance) or Negative performances (4.0%, where the migrants have obtained poor performance). The positive performances were mainly related to successful internship and training, such as "(10) Thanks to this internship, he acquires new technical and soft skills.", or finding a job, such as "(1) . . . take his CV to all construction sites in the area. This action leads him to find a job.", while it is the opposite in negative performances, such as "(1) Once the fixed-term contract is terminated, it is not renewed because, according to the employer, it has not acquired enough autonomy to continue".

Hypothesis 8 is partially confirmed in the second part since many of the experiences are limited to training operational activities usually proposed to migrants, while it is not confirmed in the first part because there are many experiences mentioned, both for training and work. However, they are conceived more as actions, which lack a reference objective, than as learning experiences.

*In the distal phase, hp. 9, refugees are not helped in capitalizing on what they can learn about themselves and their context.* Background influences have been coded as Barriers' background influences (2.0%, barriers in participating in helpful learning experiences) or Resources' background influences (3.3%, resources in participating in helpful learning experiences). References to background barriers are minimal and related to the operator's perception of being not skilled, such as "(1) the operator acknowledges that he did not have the skills to accompany him on a path of becoming aware of his professionalism", or accepted by the migrant, such as "(4) Amir meets the tutor and talks about his school career in Morocco".

Hypothesis 9 is clearly confirmed above all by the fact that it seems to be lacking, even before the action, the intention that the operator's task is also to help the refugee develop an awareness of himself and the context.

*Hp. 10, in the proximal stage, operators tend to replace migrants in choosing objectives and action strategies and to propose training and work opportunities in a prescriptive way.* Proximal

influences have been coded as Barriers' proximal influences (5.9%, influences on goals and actions choices that have not been effective or accepted by the migrants) or Resources' proximal influences (2.0%, influences on goals and actions choices that have been effective and accepted by the migrants). Proximal influences are more numerous than the background ones. Some are considered barriers because they substitute for actions that the migrants can and should manage on their own, such as "(2) the operator is applying for an internship position in a toy company that is looking for a warehouse worker.", or propose something the migrants do not want to consider, such as "(9) the operators continue to explain to Amir the importance of starting a job to be able to sustain themselves economically."; others are considered resources because their influences are accepted by the migrants or congruent with their skills and interests, such "(14) the operator offers him a course as an electrician in line with what he wants to do".

Hypothesis 10 is also mainly confirmed. The perception is that the operators tend to carry out "treatment" interventions in response to an immediate occupational need rather than support a choice process that generates satisfaction and autonomy.

In the end, even if they do not refer directly to any of the hypotheses tested, sharing the results related to limits and opportunities and personal inputs can be helpful. Limits and opportunities have been coded as Limits (3.0%, objective limits on the migrants' career choices) or Opportunities (3.3%, objective opportunities for the migrants' career development). Part of the limits was related to the recognition of political refugee status, such as "(7) Abou has also recently received a negative result from the court and will have to leave Italy.", and another part was related to difficulties in finding specific jobs or participating in specific training, such as "(3) The training to become a graphic designer are numerous, but all very expensive and neither the project nor the center in which it is hosted can bear this cost". Opportunities were mostly related to those provided by the companies, such as "(4) A company would have been available to hire him immediately even if he is in the second year of the course." and "(10) The company organizes training sessions to facilitate integration, and these experiences facilitate it to interact with people". The only reference directly related to personal inputs was Health status (0.3%): "(1) a health problem emerges: he suffers from eye glaucoma and should not lift weights, make efforts". Likely, the migratory background was never mentioned because it was implicit in everything told.

## 4. Discussion

Both the hypotheses that have been confirmed and those that have not seem to indicate a vision of the operator's role and, consequently, a mode of action, which is replicated even when refugees have different characteristics and needs. The operators tend to contribute in the final part of the process when the migrants have to identify their professional goals and choose which actions to take to achieve them. They view their role as prescriptive and substitutive, with the idea that they have to "take care" of needy people. Instead, they do not see their role as supportive in self-awareness, contextual knowledge, and career choice process, based on which is the idea that the person has potential and that they must be helped to recognize it and capitalize on it in the new context. This mode can produce immediate actions of finding employment, often of low quality, or adverse reactions in the qualified migrants, rather than developing interests and projections towards a career that creates satisfaction.

Going into more detail, it should first be emphasized that operators' influence is concentrated at the proximal level when the migrants must choose goals and actions. It looks like the operators tend to perceive their role as prescriptive and focused on facing specific tasks or problems. In this way, the migrant is not involved in the process of self-awareness and knowledge of the limits and opportunities of the hosting country. The consequence is that migrants not only do not develop career management skills, becoming day-by-day more experienced and involved in managing their careers, but also do not understand the proposal and the limits highlighted by the operators and persist in pursuing goals that can be incongruent with their skills, aptitudes, and the opportunities

and limitations of the new context. Baran et al. (2018) explain this reaction by relying on the theory of psychological contracts (Robinson and Morrison 2000; Rousseau 1998), which claims that people develop an understanding of the terms and conditions of their membership within an organization, and such notions influence perceptions of fairness or justice: "Given the inherent cultural differences and notions about the meaning of work within the context of refugee employment, we see psychological contracts as equally relevant yet more fragile for refugees than for native employees. Namely, the potential for confusion and misunderstanding is greater for people experiencing employment in the United States for the first time." (p. 103). It is essential, before acting, that operators and migrants share the need for joint work in redefining mutual knowledge and expectations, without which they risk setting off along paths with different objectives.

One of the negative consequences is that grieving is not recognized and managed. It is more evident when the migrants already have specific goals they want to pursue because they have a consistent experience and profession developed in their original country that they expect to repeat quickly and automatically in the hosting country. "Often, these highly skilled refugees had no choice but to take jobs well below their perceived skill level because they had no alternative. According to one of the subject matter experts, refugees with higher skillsets had a much more difficult time in the transition process and were often much less happy than those making more lateral moves" (Baran et al. 2018). The acceptance by migrants that they can no longer rely on their professional is a process that takes time, and that can be accelerated and reduced in its adverse effects only if the operators know how to recognize it and accompany them toward a new vision of themselves and the opportunities of the new context.

Refugees indeed run a greater risk of not being able to identify realistic objectives, but this does not mean that the operators should make this choice in their place. Most of the migrants' goals are abandoned, partly because they are unrealistic, partly because they are not developed directly by themselves and are felt to be imposed by the outside. It is not enough to rely on trust in operators, even when they have signs of having earned it. Defining which expectations are realistic and which are not is challenging for refugees and operators (Segal and Mayadas 2005); for this reason, the task must be managed by the two together, asking for the support of experts and guaranteeing they will agree on the conclusions.

For this reason, it is crucial to involve the migrant in the process of gaining awareness of the real opportunities provided by the new context. It does not work to "inform" the hosting country's new limits and opportunities, even if they are experts in doing so: migrants need to learn by experience, mainly at the beginning when they do not know whom to rely on. Only in this way can they convince themselves of the need to consider that the situation has changed, and they need to know it before deciding what to expect and do. Therefore, the operators' role should be that of helping migrants in managing expectations in the form of a "realistic life preview", helping them develop a realistic job preview associated with satisfaction and commitment (Premack and Wanous 1985).

Fundamental, however tricky and time-consuming, are all the interventions that can develop a more complete and realistic perception of self-efficacy in one way or another. Migrants, in the same way as everybody else, need first to trust in themselves, to feel they have professional skills and aptitudes, and the ability to face the job challenges they will choose. The first task the operators should pursue is to help the migrants to recognize their skills and aptitude and review those they overestimate.

Refugees' experiences, often many, even if in some cases too limited to activities typically offered to migrants, can be particularly useful in the career choice process, as long as they are made to understand what these experiences have to teach them. In most cases, self-efficacy perceptions and outcome expectations are developed as a secondary result of experiences which were activated only to gain a job. As a result, these and previous experiences do not fully exploit what they can teach the migrants, reducing them to an instrument to convince the potential employer to hire them. Even when they

are chosen to respond to an occupational urgency, they must be offered an opportunity for knowledge, and they must be accompanied to make migrants more competent about themselves and the new context. " ... refugees base their preconceptions about what they could do for work (once resettled) upon their prior work experiences. [ ... ] Given the often-stark differences between a refugee's home country and the [hosting country], however, refugees are likely to avail themselves of other people's experiences [ ... ]. The accuracy of those experiences and inferences will likely influence their degree of understanding, frustration, or disappointment upon resettlement." (Baran et al. 2018, p. 103). Other people's experiences are a weak channel for developing awareness of the new situation, much more if they are limited to those reported by other migrants; it is much more effective to provide the migrants with direct experiences aimed at developing a more complete and realistic knowledge of the new context.

Therefore, guidance counselors must rethink their role, objectives, and methods of action. Contrary to what is now being done, operators should not only intervene close to a choice but also mainly as activators of a process that, through targeted experiences, helps the migrant acquire awareness and set challenging objectives with a reasonable probability of generating satisfaction.

The critical step is to enable refugees to develop effective interests. Migrants, when asked, usually tend not to communicate their interests. It is likely partly due to the belief that sharing a generic availability to work is more effective. However, it is partly because some have not developed interests, lacking an adequate perception of self-efficacy and realistic outcome expectations. Outcome expectations, in this way, do not have a consistent effect on the interests' development, while it is likely they directly affect both goals and actions. The inconsistent expectations, more than double that of the congruent, can explain why goals and actions loss are more frequent than those confirmed. A qualitative integration in the labor market of the hosting country is possible only to the extent that the operators no longer aim to push migrants into the labor market as quickly and economically as possible but to support them in developing interests which, based on the skills and potential of the person and the opportunities and limits of the context, have a good chance of directing the migrant towards a career full of satisfaction.

## 5. Recommendations

From the study findings, it is possible to suggest some recommendations that concern not only operators but also institutions and migrants.

As far as counselors are concerned, it is essential, as has already been said, that they change their vision of their role and relationship with refugees, not considering themselves as experts who have to replace those who are incapable. Instead, they must consider themselves more similar to traveling companions who enable the refugee to stand on his own two feet and take responsibility for his own career choices. It is easy to understand that this change of perspective requires new and specific skills, and therefore the need to train to acquire "career choices educator" skills that must integrate well with the intercultural skills they should already possess.

For this to happen, it is essential that the institutions, both public and those that make up the reception system, recognize the failure of an approach aimed exclusively at promoting immediate job placement and set themselves higher quality objectives. Among these, the first is to encourage the development of the careers of political refugees, creating more extensive skills and satisfaction. More general, but dependent on the former, is the goal of generating inclusion, not just integration, which means questioning one's way of operating according to new methods that respond to everyone's characteristics and needs (Magnano et al. 2022). The first change, because it is the most urgent, is to recognize that the operators who deal with favoring the integration of the labor market cannot be the same ones who deal with all the other needs of refugees: a specific professional figure is needed for their skills and role within the reception centers, with the condition of being

able to be recognized as a career-counseling expert who can make a difference in the lives of refugees.

Finally, it is essential that the vision that refugees themselves have of their role in the host country also changes. This transition is complicated by the migratory project, especially when this is short-term or, as often happens, changes over time. Especially in refugees who come from countries with war conflicts, there is a desire for everything to end and for them can go home. However, often this does not happen quickly, and, over time, the refugee integrates more and more into the host country to such the point of no longer feeling like organizing the return journey. If the stay is short, it is understandable that there is no intention of investing in learning the language and a specific profession, but if the situation changes over time, and even more so for those who have decided to move permanently, it is crucial to think in a prospective logic of career development and limit the idea that one's need is only to respond to an employment urgency.

## 6. Conclusions, Limitations, and Further Research

This study has provided a new picture of the reception operators' perception of their role in supporting migrants in integrating into the labor market of the hosting country. Focusing on the proximal influences that emerge from their narrations does not mean that, in daily actions, they are not flanked by supporting migrants in developing realistic self-efficacy perceptions and outcome expectations and that interests are not considered in career development. Still, it denotes a lack of awareness and attention to a process that cannot be reduced to listing training and work experiences and the search for immediate employment to encourage the development of careers that generate satisfaction and integration.

However, some limitations to be overcome with subsequent research must be considered. The fact that the main dynamics that emerged were found in most of the cases considered should not underestimate the limit of the number of cases considered. Future research should have more extensive and representative samples of the nation considered. A second limitation is that these results could be typical of some countries but not others. It can be beneficial to compare the methods of supporting labor integration in countries where this is not very effective with those achieving better results to understand what makes the most difference, and invite some institutions to change their perspective of the action. In parallel, the possibility that the results obtained here can also be extended to the female population should be verified, considering that the sample was mainly made up of men.

Limiting ourselves to the operators' stories was a strategic choice because the objective was to focus on their perception of their role, but, as mentioned above, their intervention is probably at least in part more in line with what was suggested by the SCCT.

Limiting ourselves to the operators' stories was a strategic choice because the objective was to focus on their perception of their role. However, as mentioned above, their intervention is probably at least in part more in line with what was suggested by the SCCT. Further research should be based on in-depth interviews to help reception operators share a broader and more complete report of their activities and attitudes on labor integration. In this case, the coding system developed here will probably have to be partially modified because, for example, it will no longer be necessary to indirectly deduce the reliability of the perception of self-efficacy by deducing it from having achieved (or not) a professional goal. However, it will be possible to investigate it more directly, starting from more focused narrations.

Finally, research on refugees' perceptions should complement research on operators' perceptions. This would make it easier to understand the most critical junctions, and intervene more effectively in helping them develop a different vision of their possible role in the host nation's labor market and, consequently, of the career development process that would also best guarantee their career satisfaction.

**Funding:** This research was funded by the Università Cattolica del Sacro Cuore; Research projects of interest to the University D.3.2–2018, project named "Working Out of Poverty: accompanying the poor to become dignified agents of their development".

**Institutional Review Board Statement:** The study was conducted in accordance with the Declaration of Helsinki, and did not need approval by an Ethics Committee or Institutional Review Board because the data were not collected directly from the migrants. Furthermore, they were collected in a way not to be attributable in any way to the person from the outside and hot shared in any way. To be sure that the study met the criteria of utility and methodological correctness, it was analyzed and approved by the Scientific Committee of the project ESPoR composed of university professors of psychology and law.

**Informed Consent Statement:** Informed consent was obtained from all subjects involved in the career-counseling intervention, which also contained the authorization to use the description of their case provided it was made anonymous and not attributable in any way to the person from the outside. Since their transcription, the narratives have been identified exclusively with a number.

**Data Availability Statement:** All data generated or analyzed are not publicly available due to ethical reasons.

**Acknowledgments:** Special thanks go to the reception operators who participated in the FAMI ESPoR (European Skills Portfolio for Refugees) project (PROG.2157, FAMI 2014-2020) co-financed by the Italian Ministry of the Interior (CUPJ41G18000090005).

**Conflicts of Interest:** The author declares no conflict of interest.

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
