# Peer review of "Reception Operators’ Perception of the Labor Market Integration of Refugees in Light of the Social Cognitive Career Theory"

_socsci, doi:10.3390/socsci12010019_

Round 1

Reviewer 1 Report

Research related to how refugees and asylum seekers living abroad enter the labor market would be valuable. The support they have received in their careers and how they feel about it deserve to be studied in depth.

1 The value of the study is not clearly articulated, and what are the differences between this study and existing studies? Does it raise new questions or explain new mechanisms? How significant is the role of reception operators in the employment experience of refugees?

2  Social Career Cognitive Theory provides a precise explanation of the mechanisms that shape career interests, emphasizing the role of cognitive regulation and the fact that human behavior is influenced by the interaction of self-efficacy and social processes.The purpose of this study does not seem to explain the mechanisms of individual career choice, so is it appropriate to apply this model? 

3 The introduction of The Social Cognitive Career Theory and the definition of the study variables in the manuscript do not cite any previous studies, which should be inappropriate.

4 The 14 cases interviewed ranged in age from 19 to 40, which means they were at different stages of their careers. Their career expectations and the career dilemmas they faced were quite different. Is this difference taken into account in the interpretation of the study findings?

5 The interviewees were basically men; do the findings apply to women as well?

6 Research recommendations can be considered for reception operators as well as for refugees themselves, and should be linked to research findings.

7 It is suggested to add research limitations at the end of the manuscript.

Author Response

Dear Reviewer,

after receiving the revisions, I rethought the article’s structure to make it more congruent with an expectation of greater robustness regarding the initial investigation hypothesis.

While the results are very much in line with what my collaborators and I have observed in the field, our perception does not constitute a sufficiently authoritative basis to be able to propose it as a hypothesis to be tested.

Therefore, I thought of indicating a series of hypotheses that emerge from the SCCT model. Since several steps in the choice process can invalidate the achievement of satisfactory results, I have transformed them into precise hypotheses which are largely confirmed by the data, giving strength to the main interpretative hypothesis, which in any case, remains unchanged.

Research related to how refugees and asylum seekers living abroad enter the labor market would be valuable. The support they have received in their careers and how they feel about it deserve to be studied in depth.

Thank you for acknowledging my research efforts in this field.

1 The value of the study is not clearly articulated, and what are the differences between this study and existing studies? Does it raise new questions or explain new mechanisms? How significant is the role of reception operators in the employment experience of refugees?

I have now explicitly indicated that the research’s novelty lies in using SCCT not as a general theoretical reference but as an interpretative model both for the definition of specific hypotheses and for data encoding. To their verification, referring to the methodology of the grounded theory is added the attempt to identify an operational model common to the majority of reception operators, which explains the critical issues that may emerge from the possible confirmation of the hypotheses.

2  Social Career Cognitive Theory provides a precise explanation of the mechanisms that shape career interests, emphasizing the role of cognitive regulation and the fact that human behavior is influenced by the interaction of self-efficacy and social processes. The purpose of this study does not seem to explain the mechanisms of individual career choice, so is it appropriate to apply this model?

"SCCT may be especially valuable in understanding the career development processes of recent immigrants and refugees. Personal and environmental factors significantly influence their confidence in obtaining employment and growing in a career path of their choice." (Yakushko, Watson, and Thompson 2008)

This study focuses exclusively on reception operators’ interventions as environmental factors: I agree with the need to highlight this specificity explicitly.

3 The introduction of The Social Cognitive Career Theory and the definition of the study variables in the manuscript do not cite any previous studies, which should be inappropriate.

I have now better highlighted that both the description and the graphical model are based on the original manuscript, which continues to be the main reference to describe the model.

Lent, Robert W., Steven D. Brown, and Gail Hackett. 1994. "Toward a Unifying Social Cognitive Theory of Career and Academic Interest, Choice, and Performance." Journal of Vocational Behavior 45 (1): 79-122. doi:10.1006/jvbe.1994.

4 The 14 cases interviewed ranged in age from 19 to 40, which means they were at different stages of their careers. Their career expectations and the career dilemmas they faced were quite different. Is this difference taken into account in the interpretation of the study findings?

Now I specified that on the one hand, the sample appears rather heterogeneous, which allows it to be sufficiently representative of the refugee population in Italy; on the other hand, it is deliberately united by the number of years of residence in Italy. This aspect is particularly important because it characterizes an intense and specific phase of the labor integration process because it takes place after the first years of life in the new nation, is mainly dedicated to the acculturation process, and precedes the phase of autonomy from the reception system.

This is believed to be the best condition to identify, if there are, operators' behavioral methods that do not depend on the needs and specificities of the individual migrant or specific targets but which are characterized by being a transversal method, therefore probably acting automatically to the mere fact of dealing with political refugees.

5 The interviewees were basically men; do the findings apply to women as well?

I think the women’s situation is usually much more complicated, so it could be that the operational model of reception operators is at least in part different.

I have added, in conclusion, that the possibility that the findings here obtained can also be extended to the female population should be verified.

6 Research recommendations can be considered for reception operators as well as for refugees themselves, and should be linked to research findings.

Recommendations have now been included in the manuscript.

7 It is suggested to add research limitations at the end of the manuscript.

Research limitations have now been included in the manuscript.

Reviewer 2 Report

Thanks for inviting me to review the paper “Reception operators' perception of the labor market integration of refugees in the light of the Social Cognitive Career Theory”. Some recommendations and questions are provided.

The paper is interesting is somewhat of significant, since the issue of migrant workers’ situation is of interest.

1.      Abstract – should note how many participants. “The narrations were coded according to the Social Cognitive Career Theory” – do you mean the emerging themes are based on the Social Cognitive Career Theory?

2.      The main result is that operators tend to intervene directly and late when the migrants must choose goals and actions – rephrase, sentence unclear

3.      Try to use the journal suggested citation reference style

4.      Paper needs to be grammar check for clarity and coherence

5.      Line 84 – would suggest this to be the theoretical framework or underpinning theory

6.      There are no references for the SCCT, recheck – could also provide previous studies using SCCT in relation to the current study

7.      Participants – voluntary nature of the cases used, informed consent and ethical issues. IRB approval should be noted

8.      What now? After the findings, what practical recommendations? Or policy recommendations?

In sum, the study is of interest, however, was not able to connect most of the results with the discussions, data analysis could also be clarify, the author mentioned the use NVivo, how about the emerging themes? Constant comparison? Or just by word frequency? how about the validity and reliability?

Author Response

Dear Reviewer,

after receiving the revisions, I rethought the article’s structure to make it more congruent with an expectation of greater robustness regarding the initial investigation hypothesis.

While the results are very much in line with what my collaborators and I have observed in the field, our perception does not constitute a sufficiently authoritative basis to be able to propose it as a hypothesis to be tested.

Therefore, I thought of indicating a series of hypotheses that emerge from the SCCT model. Since several steps in the choice process can invalidate the achievement of satisfactory results, I have transformed them into precise hypotheses which are largely confirmed by the data, giving strength to the main interpretative hypothesis, which in any case, remains unchanged.

The paper is interesting is somewhat of significant, since the issue of migrant workers’ situation is of interest.

Thank you for acknowledging my research efforts in this field.

1.      Abstract – should note how many participants. “The narrations were coded according to the Social Cognitive Career Theory” – do you mean the emerging themes are based on the Social Cognitive Career Theory?

The Social Cognitive Career Theory has been the theoretical reference both to define the hypotheses to be tested and the coding of the narratives. This and the number of participants has been included in the abstract.

2.      The main result is that operators tend to intervene directly and late when the migrants must choose goals and actions – rephrase, sentence unclear

The abstract has been consistently revised: I hope now it is more clear and complete.

3.      Try to use the journal suggested citation reference style

I am using RefWorks selecting Chicago 17th Edition. I cannot find the style proposed by the journal.

4.      Paper needs to be grammar check for clarity and coherence

5.      Line 84 – would suggest this to be the theoretical framework or underpinning theory

I have made numerous changes in the manuscript so that the role of the SCCT in the study should now be clear.

6.      There are no references for the SCCT, recheck – could also provide previous studies using SCCT in relation to the current study

I have now better highlighted that both the description and the graphical model are based on the original manuscript, which continues to be the main reference to describe the model.

Lent, Robert W., Steven D. Brown, and Gail Hackett. 1994. "Toward a Unifying Social Cognitive Theory of Career and Academic Interest, Choice, and Performance." Journal of Vocational Behavior 45 (1): 79-122. doi:10.1006/jvbe.1994.

7.      Participants – voluntary nature of the cases used, informed consent and ethical issues. IRB approval should be noted

These parts have been included now.

8.      What now? After the findings, what practical recommendations? Or policy recommendations?

Recommendations have now been included in the manuscript.

In sum, the study is of interest, however, was not able to connect most of the results with the discussions, data analysis could also be clarify, the author mentioned the use NVivo, how about the emerging themes? Constant comparison? Or just by word frequency? how about the validity and reliability?

In this version of the manuscript, results are clearly linked with hypotheses designed according to the SCCT. To their verification, referring to the methodology of the grounded theory is added the attempt to identify an operational model common to most reception operators, which could explain the critical issues that may emerge from the possible confirmation of the hypotheses.

Round 2

Reviewer 1 Report

The manuscript has been improved and can be considered for publication in Social Sciences.

Reviewer 2 Report

Author/s already revised the paper according to the previous recommendations and inquiries. Just a minor technical issue: for percentages presented within the article use "." for decimal point instead of ","